Palaeoecology of the Hiraiso Formation (Miyagi Prefecture, Japan) and implications for the recovery following the end-Permian mass extinction

Foster William J. 1 2 w.j.foster@gmx.co.uk
Godbold Amanda 3 4
Brayard Arnaud 5
http://orcid.org/0000-0001-8444-8440 Frank Anja B. 1
Grasby Stephen E. 6
Twitchett Richard J. 7
Oji Tatsuo 2
1 Institute for Geology, Universität Hamburg , Hamburg , Germany
2 Nagoya University , Nagoya , Japan
3 University of Southern California , Los Angeles , United States
4 University of Tokyo , Tokyo , Japan
5 Université de Bourgogne , Bourgogne , France
6 Geological Survey of Canada , Calgary , Canada
7 Natural History Museum , London , United Kingdom
De Baets Kenneth
Electronic publication date: 2022 Dec 19
Publication date: 2022
Volume: 10
Electronic Location ID: e14357
Received 2022 Mar 18; Accepted 2022 Oct 18
Copyright: © 2022 Foster et al.
Copyright year: 2022
Copyright holder: Foster et al.
License: This is an open access article distributed under the terms of the Creative Commons Attribution License, which permits unrestricted use, distribution, reproduction and adaptation in any medium and for any purpose provided that it is properly attributed. For attribution, the original author(s), title, publication source (PeerJ) and either DOI or URL of the article must be cited.
License URL: https://creativecommons.org/licenses/by/4.0/

Keywords: End-Permian extinction, Recovery, Anoxia, Spathian, Trace fossils

Funding: British Council and JSPS Summer Programme Nagoya University Museum funded subsequent fieldwork Mitacs-JSPS Summer Program Grant NERC Grant NE/I005641/2 This study was funded by a British Council and JSPS Summer Programme grant to William Foster in 2014, Nagoya University Museum funded subsequent fieldwork for William Foster in 2016, Amanda Godbold was funded by the Mitacs-JSPS Summer Program grant, and Richard Twitchett was funded by the NERC grant (NE/I005641/2). The funders had no role in study design, data collection and analysis, decision to publish, or preparation of the manuscript.

==============================
The Hiraiso Formation of northeast Japan represents an important and under-explored archive of Early Triassic marine ecosystems. Here, we present a palaeoecological analysis of its benthic faunas in order to explore the temporal and spatial variations of diversity, ecological structure and taxonomic composition. In addition, we utilise redox proxies to make inferences about the redox state of the depositional environments. We then use this data to explore the pace of recovery in the Early Triassic, and the habitable zone hypothesis, where wave aerated marine environments are thought to represent an oxygenated refuge. The age of the Hiraiso Formation is equivocal due to the lack of key biostratigraphical index fossils, but new ammonoid finds in this study support an early Spathian age. The ichnofossils from the Hiraiso Formation show an onshore-offshore trend with high diversity and relatively large faunas in offshore transition settings and a low diversity of small ichnofossils in basinal settings. The body fossils do not, however, record either spatial or temporal changes, because the shell beds represent allochthonous assemblages due to wave reworking. The dominance of small burrow sizes, presence of key taxa including Thalassinoides, Rhizocorallium and Holocrinus, presence of complex trace fossils, and both erect and deep infaunal tiering organisms suggests that the benthic fauna represents an advanced stage of ecological recovery for the Early Triassic, but not full recovery. The ecological state suggests a similar level of ecological complexity to late Griesbachian and Spathian communities elsewhere, with the Spathian marking a globally important stage of recovery following the mass extinction. The onshore-offshore distribution of the benthic faunas supports the habitable zone hypothesis. This gradient is, however, also consistent with onshore-offshore ecological gradients known to be controlled by oxygen gradients in modern tropical and subtropical settings. This suggests that the habitable zone is not an oxygenated refuge that is only restricted to anoxic events. The lack of observed full recovery is likely a consequence of a persistent oxygen-limitation (dysoxic conditions), hot Early Triassic temperatures and the lack of a steep temperature/water-depth gradient within the habitable zone.

Introduction

The end-Permian mass extinction caused the greatest taxonomic and ecological changes of the Phanerozoic with approximately 90% of marine and 70% of terrestrial species appearing to have gone extinct (Erwin, 1993), reef ecosystems collapsing (Senowbari-Daryan et al. 1993; Martindale, Foster & Velledits, 2019), an increase in the proportion of Lazarus taxa (Erwin & Pan, 1996; Twitchett, 2001), the loss of the sedimentary mixed layer (Hofmann et al., 2015), a significant shift in the evolutionary trajectory of life (Muscente et al., 2018), and the global homogenisation of faunal communities (Kocsis et al., 2018). The current consensus is that rapid global warming, triggered by the emplacement the Siberian Traps Large Igneous Province was the cause of the crisis and can account for most of the features associated with the extinction event (Burgess, Muirhead & Bowring, 2017; Clapham & Renne, 2019).

The recovery following the mass extinction is also considered unusually “delayed”, as the gap between extinction and full recovery is the longest of any mass extinction and proportionally longer than would be expected for the magnitude of the biodiversity loss (Erwin, 2001). Questions concerning the pattern and duration of the recovery depends on the clades being investigated, the spatial scale being addressed, and the metrics being used to assess recovery. Ammonoids show a rapid diversification and high species richness in the Smithian (Brayard et al., 2009b), whereas bivalves and gastropods show a hyperbolic rediversification pattern with an Early Triassic delay and an explosive radiation in the Middle Triassic (Friesenbichler, Hautmann & Bucher, 2021). From the perspective of global biodiversity, however, it still took until the Jurassic for family-level marine diversity to return to pre-extinction levels, and large metazoan reefs did not reappear until the Pelsonian-Illyrian approximately 11 million years after the mass extinction (Martindale, Foster & Velledits, 2019). On the other hand, if ecological attributes of marine ecosystems such as tiering, body size, and the presence of index taxa are considered, as in semi-quantitative ecosystem-recovery models (e.g., Twitchett, 2006), then locally, fossil assemblages recording rapid recovery have been documented from some locations from late Griesbachian successions (Twitchett et al., 2004; Hofmann et al., 2011; Foster et al., 2017), and complex marine ecosystems have been recorded at some sites in the Olenekian (Brayard et al., 2011, 2017). The Early Triassic is also well-known for having a poor fossil record of skeletal invertebrates with an unusually high number of Lazarus taxa (Twitchett, 2001) confounding our understanding on the dynamics of recovery. The ichnological record has, therefore, become increasingly important in understanding the pattern of recovery following the mass extinction (Twitchett & Barras, 2004).

The pattern and duration of recovery has been recorded to vary between different regions (Twitchett & Barras, 2004; Twitchett et al., 2004), and different environmental settings (Beatty, Zonneveld & Henderson, 2008; Zonneveld, Gingras & Beatty, 2010). A number of different hypotheses have been proposed to explain the different spatial and temporal patterns of recovery. Shallow marine ecosystems are thought to have recovered faster at higher latitudes (Wignall, Morante & Newton, 1998; Twitchett & Barras, 2004; Beatty, Zonneveld & Henderson, 2008), although this idea has been challenged (Fraiser & Bottjer, 2009; Knaust, 2010; Hofmann et al., 2011). It has also been proposed that shallow marine settings above wave base provided oxygenated refugia that allowed marine ecosystems to recover faster than in deeper marine settings that had become anoxic/euxinic, i.e., the habitable zone hypothesis (Beatty, Zonneveld & Henderson, 2008; Zonneveld, Gingras & Beatty, 2010). Whilst data from many sections do support the habitable zone hypothesis (Godbold et al., 2017; Foster et al., 2017; Woods et al., 2019), not all sections above wave base record rapid recovery following the mass extinction (Twitchett et al., 2004; Foster et al., 2015; Feng et al., 2017). Investigations of the δ18O suggests that there were generally high temperatures throughout the Early Triassic, but with a highly dynamic climate with subsequent warming and cooling phases associated with other environmental changes (Sun et al., 2012; Romano et al., 2012). This led Song et al. (2014) to suggest that the high temperatures that characterise the Early Triassic also restricted recovery from the shallowest environments suggesting that the habitable zone is above wave base but also below lethally hot temperatures.

Associated with these environmental changes it has also been shown that the pace of recovery for clades and faunal communities that do recover during the Early Triassic were setback by subsequent biotic crises (Ware et al., 2011; Foster et al., 2017; Wu et al., 2019). A key event is the hypothesised “late Smithian Thermal Maximum” that coincides with the highest temperatures reconstructed for the Early Triassic and a subsequent cooling interval that characterises the early Spathian (Sun et al., 2012; Goudemand et al., 2019). This event is also characterised by a re-expansion of oxygen minimum zones into shallow marine settings (Grasby et al., 2013; Sun et al., 2015; Wignall et al., 2016; Zhao et al., 2020). The trigger of the “late Smithian Thermal Maximum” or the other subsequent climate crises is, however, uncertain. Some researchers suggest that subsequent volcanic pulses from the Siberian Traps is the underlying cause (Payne & Kump, 2007), whereas others demonstrate that the role of volcanism is equivocal (Hammer et al., 2019). Either way, this event is linked to a significant decline in diversity of conodonts and ammonoids (Brayard et al., 2006; Stanley, 2009; Wu et al., 2019) and a turnover in benthic marine ecosystems (Hofmann et al., 2014; Hofmann, Hautmann & Bucher, 2015; Foster et al., 2015, 2017, 2018a). The Triassic recovery of marine ecosystems is, therefore, the recovery from not just the end-Permian mass extinction, but also multiple climate crises throughout the Early Triassic.

Our understanding of the pattern and duration of recovery for benthic marine ecosystems is limited to only a few regions and to only a few localities within those regions, in particular the western United States (Schubert & Bottjer, 1995; McGowan, Smith & Taylor, 2009; Brayard et al., 2011, 2015, 2017; Hofmann et al., 2014; Pietsch, Mata & Bottjer, 2014), central Europe (Hofmann, Hautmann & Bucher, 2015; Foster et al., 2015, 2017, 2018a; Pietsch, Petsios & Bottjer, 2016) and South China (Payne et al., 2006, 2011; Chen et al., 2007; Chen, Tong & Fraiser, 2011; Hautmann et al., 2015; Foster et al., 2018b; Feng et al., 2018). The main aim of this study is, therefore, to quantify the recovery of benthic invertebrates during the Early Triassic from the previously underrepresented Kitakami Massif, northeast Japan. The Permian-Triassic successions from Japan are well-known for their bedded chert successions that preserve pelagic deep-sea facies from the ocean floor to the lower flanks of mid-Panthalassan seamounts (Isozaki, 1997; Onoue, Soda & Isozaki, 2021), and carbonate successions that preserve deposition on mid-Panthalassan seamounts (Sano & Nakishima, 1997). The Japanese successions also record clastic successions of shallow marine basins from the western margin of Panthalassa (Kashiyama & Oji, 2004; Yoshizawa et al., 2021). Even though there has been extensive research and systematic surveys of Permian-Triassic faunas and established biostratigraphic frameworks (Nakazawa, 1953, 1959; Hayami, 1975; Ehiro, Sasaki & Kano, 2016) there are very few palaeoecological studies of the Japanese successions.

To elucidate the pattern and duration of recovery between different regions after the mass extinction, we undertook a quantitative palaeoecological study of the Hiraiso Formation, northeast Japan, which is a shallow marine clastic succession. Here we collected new data to resolve the stratigraphical framework, and investigate the diversity, composition and size of both trace fossils and benthic invertebrates to understand the pace and pattern of recovery. In addition, we utilised geochemical proxies to understand how these ecological changes related to redox conditions (using V/Cr and Th/U). We then compare the ecological state of the benthic fauna of the Hiraiso Formation with coeval assemblages from around the world to investigate the similarities and differences in the pattern and duration of recovery from the end-Permian mass extinction.

Geological setting

Japan is an accretion complex in which many terranes of several origins are juxtaposed and several allochthonous blocks are involved. Unlike the well-known Early Triassic carbonate succession from the Chichibu terrane in southern Japan, and the pelagic chert-claystone succession from the Mino-Tamba terranes in central Japan, which are interpreted as seamount and deep sea settings from central Panthalassa respectively, the South Kitakami terrane is a siliciclastic succession that was located in the western Panthalassa Ocean (near Primorye) in the Lopingian and Early Triassic (Sano & Nakishima, 1997; Isozaki, 1997; Kobayashi, 1999, 2002; Brayard et al., 2009a; Shigeta, 2021) (Fig. 1). The palaeoposition of the South Kitakami terrane during the Permian-Triassic transition is, however, a matter of debate (see Shigeta, 2021). Most recently, Isozaki et al. (2017) suggested that the South Kitakami terrane was located at the northeast tip of the South China block, and that both South Primorye and the South Kitakami terrane belonged to “Greater South China”.

Figure 1 A summary map of the basement geology of Japan showing the terranes that have sedimentary sequences that include Triassic rocks, with an inset map showing the palaeogeographic location of the South Kitakami terrane during the Triassic (red circle).

Ch, Chichubu terrane, Ks, Kurosegawa terrane, Sa, Sangun terrane, Mz, Maizuru terrane, Jo, Joetsu terrane, Nk, North Kitakami-Oshima terrane, Id, Idonnappu terrane. Modified from Wallis et al. (2020). TTL, Tanakura Fault. ISTL, Itoigawa-Shizuoka Fault. Blue lines represent plate boundaries. Palaeogeographic map after Blakey (2012).

The Triassic succession of the South Kitakami terrane belongs to the Inai Group which can be divided into: the Hiraiso, Osawa, Fukkoshi, and Isatomae formations. The Hiraiso Formation unconformably overlies the Permian (Wuchiapingian-Changhsingian) Toyoma Formation, and the Hiraiso Formation has been correlated with the Olenekian stage (Nakazawa et al., 1994; Shigeta & Nakajima, 2017; Shigeta, 2021). The end-Permian mass extinction and Permian/Triassic boundary are not preserved on the South Kitakami terrane. At its base, the Hiraiso Formation is a conglomerate overlain by a 250 m-thick succession of sandstones intercalated with siltstones (Kamada, 1989; Ehiro et al., 2016). Tempestites with abundant disarticulated bivalves and crinoids are frequent in the succession, and the Hiraiso Formation has been interpreted to represent deposition on a storm-influenced ramp (Kamada & Kawamura, 1988; Kashiyama & Oji, 2004). The overlying Osawa Formation, on the other hand, is dominated by siltstones and mudstones (Kamada, 1989) with abundant ammonoids (Bando & Shimoyama, 1974; Ehiro, 1993), thylacocephalans (Ehiro et al., 2016), coprolites (Nakajima & Izumi, 2014), and marine reptile fossils (e.g., Utatsusaurus hataii (Tokio & Masafumi, 1978)), which represents deposition in a low-energy basinal setting. The Hiraiso Formation transitions into the Osawa Formation and a specific horizon to draw the boundary between the two formations is difficult to discern. The base of the Osawa Formation is, therefore, distinguished from the Hiraiso Formation by the absence of macroscopic bioturbation and benthic invertebrate fossils.

The age of the Hiraiso Formation is poorly constrained due to the lack of stratigraphic index fossils, i.e., conodonts are unrecorded and previously only three ammonoid specimens have been described. One ammonoid specimen originally identified as Glyptophiceras cf. gracile led Bando (1970) to assign the Hiraiso Formation as Griesbachian. However, Nakazawa et al. (1994) re-identified the specimen as G. aequicostatus and reassigned the Hiraiso Formation to the Smithian. Subsequently, Shigeta (2021) re-identified the specimen as the lower Spathian species Deweveria kovalenkoi. Furthermore, provenance of this specimen was uncertain as it was collected from float material, and Shigeta (2021) provides evidence for an occurrence in the lower part of the Osawa Formation instead of the Hiraiso Formation. Shigeta & Nakajima (2017) recently extracted an ammonoid fragment from a massive sandstone near the base of the Hiraiso Formation, which they identified as a lower Spathian species Tirolites cf. ussuriensis. This led Shigeta & Nakajima (2017) to reassign the Hiraiso Formation to the lower Spathian following a correlation with South Primorye. The overlying Osawa Formation, on the other hand, has a richer and more abundant ammonoid fauna that correlates with the Spathian (Bando & Shimoyama, 1974; Ehiro, 1993; Ehiro, Sasaki & Kano, 2016). A revision of the described ammonoids from the Osawa Formation led Shigeta (2021) to interpret a correlation with the Neocolumbites insignis ammonoid zone of South Primorye, which represents deposition in the upper part of the lower Spathian. This age assignment, therefore, restricts deposition of the Hiraiso Formation to the lower Spathian. In this study, we collected two new ammonoid specimens from the lower part of the Hiraiso Formation, and the presence of Koninckitoides supports a lower Spathian age (see Results and Discussion). Other ammonoids were also found from the Hiraiso Formation but could not be prepared for taxonomic identification.

Materials and Methods

The Hiraiso and Osawa formations were investigated at thirteen exposures between the Maekawara and Akaushi ports, an exposure at Kamiwarizaki, and Osawa Port (Figs. 3, A1–A4, Table 1). Combined, the exposures between the Maekawara and Akaushi ports provide an almost continuous exposure of the Hiraiso Formation. Fieldwork to these locations was registered with the Ministry of the Environment, Tohoku Regional Environment Office. Lithologies, sedimentary structures, ichnofabric indices (Droser & Bottjer, 1986), and trace fossils (including burrow size and tiering depth) were described for each measured bed. Bivalves on bedding surfaces were measured and identified at species-level where possible. Fossils were measured in the field (minimum size 0.5 mm) with the height and length of each specimen measured using digital calipers to the nearest 0.1 mm. The geometric mean of each specimen was calculated following Jablonski (1996): square root of the product of shell height and shell length. Both burrow and bivalves sizes were loge transformed for lithofacies comparisons so that proportional deviations are represented consistently (Kerkhoff & Enquist, 2009).

Table 1 Locations of each of the investigated sections.

Section	Base of section	Top of section	
Hiraiso Coast 1a	38°48′0.30″N, 141°32′45.89″E	38°48′1.09″N, 141°32′45.61″E	
Hiraiso Coast 1b	38°48′0.27″N, 141°32′44.87″E	38°48′0.48″N, 141°32′44.65″E	
Hiraiso Coast 2	38°47′49.61″N, 141°32′38.53″E	38°47′50.00″N, 141°32′36.55″E	
Hiraiso Coast 3	38°47′49.20″N, 141°32′35.20″E	38°47′51.10″N, 141°32′33.57″E	
Hiraiso Coast 4	38°47′51.35″N, 141°32′32.47″E	38°47′51.31″N, 141°32′26.57″E	
Hiraiso Coast 5	38°47′51.63″N, 141°32’26.16″E	38°47′53.35″N, 141°32′25.26″E	
Hiraiso Coast 6	38°47′53.61″N, 141°32′23.38″E	38°47′53.74″N, 141°32′22.59″E	
Osawa Harbour	38°47′30.76″N, 141°31′43.46″E	NA	
Kamiwarizaki	38°37′41.70″N, 141°31′40.92″E	38°37′41.74″N, 141°31′40.10″E	

Beds that yielded body fossils were well-cemented and too hard for effective mechanical disaggregation. These beds with shelly fossils were, therefore, polished to reveal the fauna. In addition, samples were collected every 1 m where possible for ichnofabric analysis. For the body fossils, a 5 × 5 cm quadrat was placed over the polished slab of each sample and all bioclasts within this area were counted to obtain abundance and taxonomic composition data. All identifiable fossils in the polished slabs were identified to the most precise taxonomic level to which they could be confidently assigned (Supplement S5). The polished slabs that were not shell lenses were investigated for their ichnofabric, where: trace fossils were identified to ichnogenus-level, cross-cutting relationships, ichnofabric indices (Droser & Bottjer, 1986), and tiering depth were described for each sample. Samples were then grouped into different ichnofabrics following their similarity relating to these parameters.

A cluster analysis (Q-mode) was applied to recognise those body fossil taxa that tend to co-occur in polished slab samples and to group together samples of similar taxonomic composition using the Bray–Curtis dissimilarity matrix (Clarke & Warwick, 2001). Prior to multivariate analysis, a percent transformation within samples was performed, followed by a log transformation to reduce the influence of the most abundant taxa. The similarity profile test (SIMPROF) (Clarke & Warwick, 2001) was applied to determine significant differences between the clusters. Here, 999 permutations were applied to calculate a mean similarity profile and the chosen significance level was 0.05. These clusters where then interpreted to represent different biofacies. A second cluster analysis (R-mode) was then applied to show the groups of taxa that tend to co-occur. A cluster analysis was also done for trace fossil communities recognised from bedding planes. The cluster analysis and plots were done using the software PRIMER 7 (Clarke & Warwick, 2001).

There were 118 samples covering sections 1 to 5 along the Hiraiso Coastline and all exposed facies were analysed for major, minor and trace element concentrations at the Geological Survey of Canada, Calgary. A homogenised bulk 200 mesh powder was produced using an automated agate mortar and pestle. Splits from the homogenised powder where used for subsequent analyses. Elemental determinations were conducted on powdered samples digested in a 2:2:1:1 solution of H2O-HF-HClO4-HNO3, and subsequently analysed using a PerkinElmer Elan 9000 mass spectrometer, with ±2% analytical error. Hg was measured by LECOR AMA254 mercury analyser ( ±10%). The complete data set is available in Supplement S6.

Results

The six sections logged along the Hiraiso Coastline provide a quasi-complete exposure of the Hiraiso Formation. Large faults transect the exposures at section 1 and between sections 1 and 2 (Fig. 2); it is, therefore, unknown how much of the succession is missing between these sections. Based on the similar lithofacies observed at both sections we assume that the stratigraphic gap is minimal. A gap in the exposure between sections 3 and 4 occurs in a small bay where a river meets the coastline, but no faulting was observed. The Hiraiso Formation lacks marker beds, and it was not possible to correlate the exposure at Kamiwarizaki to the Hiraiso coastline. The exposure at Kamiwarizaki is characterised by a high abundance of fossil plant material and lower shoreface lithofacies, which is a characteristic most similar to the base of the Hiraiso Formation at section 1 of the Hiraiso Coastline. The Kamiwarizaki section is, therefore, interpreted to represent the lower part of the Hiraiso Formation.

Figure 2 Locality map for the Kamiwarizaki and Hiraiso Coast sections, Miyagi Prefecture, Japan.

(A) Inset showing the location of the different sections in relation to Motoyoshi and Kesennuma in Miyagi prefecture. (B) Shows the location of the investigated section at Kamiwarizaki (highlighted in blue with a K). (C) Shows the location of the investigated sections on the Hiraiso Coast (highlighted in blue). Only the major fault lines are shown following Kashiyama & Oji (2004).

Figure 3 Field photographs showing the nature of the exposures along the Hiraiso Coast and at Kamiwarizaki.

(A) Upper Permian Toyoma Formation, Maehama Fishing Port. Hammer for scale. (B) Lower Triassic Hiraiso Formation, Hiraiso Coast, Maekawara Fishing Port. (C) Lower Triassic Hiraiso Formation, Kamiwarizaki, Kotaki Fishing Port. Person for scale. (D) Transition between the Hiraiso (exposure on the right) and Osawa Formation (exposure of the left), Akaushi Fishing Port.

Depositional setting

We identified seven lithofacies from the Hiraiso Formation that are interpreted to represent deposition from an upper shoreface to distal offshore shallow marine setting (Table 2; Fig. 4). Six of these lithofacies correspond to the lithofacies recognised by Kamada & Kawamura (1988) and Kashiyama & Oji (2004). We also recognised an additional lithofacies, which would have been included within lithofacies Ha of Kashiyama & Oji (2004). We, therefore, distinguish beds that are hummocky cross-stratified with erosive bases and concentrations of bivalve on the base, from groups of beds that are cross-stratified, and have thick sandstone beds with perpendicularly orientated large bivalves (Eumorphotis iwanowi), with the latter facies being deposited in a shallower part of the lower shoreface setting (Table 2; Fig. 4).

Table 2 Sedimentary facies and depositional environments for the investigated Hiraiso Formation.

Facies labels correspond to Kashiyama & Oji (2004), Facies Ha′ was, however, not recognised by those authors and is added in this study.

Facies	Description	Depositional environment	
S	Well-sorted, massive, thick-bedded, light grey, sandstones.	Upper shoreface	
Ha′	Swaley, cross-bedded, fine grained, thick sandstones and very fine-grained, sandstones, separated by thin (mm-size), dark grey, silt laminations. The thicker, fine-grained sandstone beds frequently contain large (up to 7 cm) bivalves orientated perpendicular to bedding.	Lower shoreface	
Ha	Hummocky cross-stratified, thick, fine-grained sandstones with randomly orientated invertebrates forming basal lenses. Sandstones alternate with thin beds of bioturbated siltstone (mm-scale).	Lower shoreface-offshore transition	
Hb	Hummocky cross-stratified, thick, fine-grained, occasionally graded upward sandstones with randomly orientated invertebrates and rip up mud clasts forming basal lenses.
Sandstones alternate with thick beds of bioturbated, dark grey, siltstones. Plant debris can be common on sandstone bedding surfaces.	Offshore transition	
Hc	Parallel- and hummocky cross-stratified, sandstones that occasionally exhibit upward grading. Asymmetrical ripples common on bedding surfaces. Sandstones also exhibit convolute bedding. Sandstones separated by beds of bioturbated, dark grey siltstones.	Offshore transition	
Hd	Parallel laminated and massive sandstones alternating with siltstones. Both sandstone and siltstone beds are typically bioturbated, but thicker sandstone beds are only bioturbated towards the bedding surface. Sandstones also exhibit asymmetrical ripples.	Offshore transition-distal offshore	
He	Banded alternations of sandstones and bioturbated siltstones. Sandstones commonly exhibit asymmetrical ripples on bedding surfaces or convolute bedding.	Distal offshore	
Ms (Osawa Fm.)	Banded alternations of unbioturbated grey sandstones and dark grey siltstones.
Sandstones typically have surfaces with small asymmetrical ripples.	Basin	

Figure 4 Schematic diagram showing a transect of the depositional environments of the Hiraiso Formation.

Estimated bathymetric ranges of the lithofacies is shown below the schematic. Depositional schematic modified from Kamada & Kawamura (1988).

Overall the Hiraiso Formation records a gradual drowning of the shelf (Figs. A1–A4); where sections 1 and 2 of the Hiraiso coastline dominantly record lithofacies Ha and Hb representing deposition in a lower shoreface and offshore transition setting; sections 3 and 4 dominantly record lithofacies Hc and Hd representing deposition around mean storm weather wave base in an offshore transition and distal offshore setting; section 5 is then represented by lithofacies Hd overlain by He representing deposition below mean storm weather wave base; and section 6, which is the base of the Osawa Formation, records a striped alternation of mudstones and sandstones that are interpreted as deposition in a basin setting. As you move up section across the Hiraiso/Osawa Formation boundary, overall the sandstone beds get thinner suggesting that the Osawa Formation Ms facies developed due to the deepening of the shelf, rather than the development of a restricted setting. The Osawa Formation also records relatively thick sandstone beds (up to 63 mm), which likely represent turbidity flows into the basin.

Systematic Palaeontology of Ammonoids

by Arnaud Brayard

All specimens are housed in the collection of National Museum of Nature and Science (NMNS), Tsukuba, Japan.

Order Ceratitida Hyatt, 1884

Superfamily Xenodiscoidea Frech, 1902

Family Proptychitidae Waagen, 1895

Genus Koninckitoides (Dagys & Ermakova, 1988)

Type species. Koninckites posterius Popov, 1961, from the early Spathian Boreoceras demokidovi beds (Bajarunia euomphala Zone) of the Lena river delta, Siberia.

Included species. Koninckites posterius Popov, 1961; Koninckitoides? taimyrensis Dagys & Ermakova, 1988; Koninckitoides solus Zakharov & Smyshlyaeva, 2016.

Diagnosis (emended after Popov (1961) and Dagys & Ermakova (1988)). Discoidal, compressed and involute proptychitid with a subtabulate to arched venter, subparallel to slightly convex inner flanks gradually converging towards venter and bearing barely perceptible to low, dense, weakly sinuous ribs weakening towards venter. Umbilical margins sharply rounded. Suture line typical of proptychitids but exhibiting a long and complex auxiliary series with relatively deep notches. Lobes are deeply indented and saddles are elongated. The second lateral saddle is often slightly bent towards umbilicus. Its umbilical flank appears weakly indented at its base.

Occurrence. This genus was previously reported from the early Spathian Bajarunia euomphala, Nordophiceras contrarium Zones and early-middle Spathian Parasibirites grambergi Zone of Siberia (Popov, 1961; Ermakova, 1981; Dagys & Ermakova, 1988), the early Spathian Columbites Zone of southeastern Idaho (Kummel, 1969), and the early Spathian Tirolites-Amphistephanites Zone of South Primorye (Zakharov & Smyshlyaeva, 2016).

Discussion. Koninckitoides is reported from a few, but distant, early Spathian localities indicating that it had a relatively cosmopolitan distribution during this time interval. Specimens were generally sampled in low abundance suggesting that it may have been present in other localities but not found or not identified so far. This low abundance also prevents a detailed knowledge of the ontogenetic and intraspecific variation of each described species. The suture line of the holotype was poorly illustrated by Popov (1961) and improperly copied by Kummel (1969). However, illustrated suture lines for the type species by Dagys & Ermakova (1988) allow detailed comparisons with reported species worldwide and the sampled specimen from the Hiraiso Formation. The structure of the suture line is characteristic with a long and complex auxiliary series showing relatively deep notches. This is rarely found in other proptychitid genera and shows what can be found for instance in some hedenstroemiids. However, the suture line also exhibits deeply indented lobes and a lateral saddle slightly bent toward the umbilicus with weakly indented flanks at their base, which are more characteristic of proptychitids. As suggested by Kummel (1969), such suture line structure and the discoidal, involute morphology may also recall the Smithian Pseudaspidites genus. However, Koninckitoides can be distinguished by its long and complex auxiliary series. The specimen NMNS PM35937 is, therefore, attributed to this genus.

Koninckitoides aff. posterius (Popov, 1961)

Figures 5A–5C

Figure 5 Ammonoids identified from the Hiraiso Formation.

(A–C) Koninckitoides aff. posterius, specimen NMNS PM35937, Hiraiso Formation, early Spathian; (C) suture line of NMNS PM35937 at H = 19 mm; (D–H) Proptychitidae gen. et sp. indet., specimen NMNS PM35938, Hiraiso Formation, early Spathian; (H) suture line of NMNS PM35938 at H = 9 mm. Scale bars: 10 mm (A and B, D–G), 5 mm (C and H).

Holotype of K. posterius. Specimen 415/6399, from the early Spathian Boreoceras demokidovi beds (Bajarunia euomphala Zone) of the Lena river delta, Siberia (Popov, 1961).

Occurrence of K. posterius. Early Spathian beds of Siberia (Popov, 1961; Ermakova, 1981; Dagys & Ermakova, 1988), Hiraiso Formation of Japan (this work), and potentially from early Spathian Columbites beds of southeastern Idaho (Kummel, 1969).

Description (specimen NMNS PM35937). Specimen with a discoidal, compressed and rather involute coiling. Venter is narrowly arched with no distinct shoulders. Flanks are subparallel near the umbilicus and become convex towards the venter. Flanks exhibit low, barely perceptible and weakly sinuous ribs fading towards the venter. Umbilicus is rather narrow with a sharply rounded margin. Umbilical wall apparently vertical. Suture line typical of proptychitids with elongated and deeply indented saddles. The second lateral saddle is slightly bent towards the umbilicus and its umbilical flank is slightly indented at its base. The suture line also bears a very elongated an complex auxialiary series with relatively deep notches. Ventral lobe is relatively large.

Materials. A single sampled specimen (NMNS PM35937).

Measurements. Not possible on the sampled specimen.

Remarks. The specimen NMNS PM35937 from the Hiraiso Formation well fits with described diagnostic characters for the Koninckitoides type of species. Similarly to K. posterius, umbilical wall of PM35937 is apparently vertical with a sharply rounded shoulder. Overall coiling parameters and ornamentation are also similar. The main difference between PM35937 and some specimens of the type series is the suture line, which apparently shows more rounded saddles for the holotype (but at a larger height size). Ermakova (1981) and Dagys & Ermakova (1988) both described the intraspecific (and intra-specimen) variation of the suture line as rather large, from rounded to elongated (“tongue shape”). This brings some doubt about the roundness of saddles for all suture lines, for all specimens and at all sizes. The illustration Text-fig. 2“u” of Dagys & Ermakova (1988) is for instance rather close to the one of PM35937. Overall, PM35937 is close in shell shape, ornamentation and suture line architecture, to K. posterius. However, additional specimens are needed to clearly redefined all reported Koninckitoides species worldwide. We thus assign PM35937 to as Koninckitoides aff. posterius, leaving the possibility that it may be a new Koninckitoides species if the small differences observed on suture lines are confirmed. Following opinion of Dagys & Ermakova (1988), K. dolosus Ermakova, 1981 is likely a more ornamented variant of K. posterius taking into account the intraspecific ornamentation variability visible on specimens of K. posterius. This is very likely based on their identical overall coiling parameters and suture line architecture. We thus considered K. dolosus as a junior synonym of the type species. Koninckitoides? taimyrensis described by Dagys & Ermakova (1988) from Siberia is potentially more involute but more material from Siberia is required to solve this question. Pseudaspidites popovi reported from southeastern Idaho by Kummel (1969) exhibits a relatively similar morphology and a highly resembling suture line (excepting apparently more spaced notches in the auxiliary series). It can therefore be conspecific with the K. posterius. However, material illustrated by Kummel (1969) is very restricted and additional representatives were not retrieved during recent exhaustive fieldworks in coeval beds from neighbouring localities in southeastern Idaho (Guex et al., 2010; Jenks et al., 2013). Grossly morphologically-close taxa such as Arctomeekoceras occur in this area but always lack the typical, long and complex auxiliary series of Koninckitoides. In this case too, new additional specimens are required to decipher the taxonomic affinities of the specimens illustrated by Kummel (1969). Koninckitoides solus described by Zakharov & Smyshlyaeva (2016) from South Primorye exhibits a somewhat similar suture line, but it mainly differs by its more involute coiling.

Proptychitid gen. et sp. indet.

Figures 5D–5H

Description. Moderately involute, compressed shell with an ogival section and convex flanks converging towards an arched venter. Ornamentation consist of marked, irregularly spaced, radial, triangular ribs, but asymmetric between the two flanks. Suture line exhibiting elongated lateral saddles slightly bent towards the umbilicus and somewhat phylloid, deeply indented lobes and a short auxiliary series with deep notches.

Material. A single sampled specimen (NMNS PM35938).

Measurements. Not possible on the sampled specimen.

Remarks. The observed suture line is characteristic of proptychitids. However, the small size of the single sampled specimen, as well as the observed asymmetric ornamentation, maybe resulting from an injury on the right flank, prevent a firm taxonomic assignment.

Ichnology

Trace fossils are commonly exposed on bedding planes of the Hiraiso Formation. In addition, where the rocks are being actively eroded by waves, the sedimentary structures (including bioturbation) can be observed clearly. Using bedding plane analysis and ichnofabric analysis, wrinkle marks and seventeen different ichnospecies were recognised (Table 3; Fig. 6). The bedding planes recorded between 1 to 6 ichnogenera and most bedding planes only recorded 1 or 2 ichnogenera (Fig. 7). The most abundant ichnogenera observed on the bedding planes (Arenicolites statheri, Diplocraterion parallelum, Thalassinoides suevicus, Rhizocorallium irregulare, Chondrites intricatus, and Teichichnus ispp.) show an onshore-offshore gradient in their distribution (Fig. 7). In lithofacies Ha, Hb, and Hc Arenicolites statheri dominates most of the bedding planes, this ichnogenus also extends into lithofacies Hd but is not the dominant ichnospecies. Thalassinoides suevicus is abundant in the shallower lithofacies Hc and the shallower samples in liothofacies Hd. Diplocraterion parallelum ranges from lithofacies Ha to He, but it dominates bedding planes in lithofacies Hd in particular when this lithofacies occurs at lower log heights, i.e., a shallower part of lithofacies Hd. As you move up-section within lithofacies Hd, Rhizocorallium irregulare becomes increasingly dominant, and dominates bedding surfaces within lithofacies Hd. The trace fossil Teichichnus ispp. also becomes more abundant on bedding planes. In the highest parts of the section in lithofacies Hd and He, Chondrites intricatus then becomes increasingly dominant.

Table 3 Descriptions of the different ichnospecies identified in this study.

Ichnospecies	Short description	Ethology	
Skolithos linearis
Haldeman 1840	Straight, vertical, cylindrical burrows. Distinct burrow wall.	Domichnia	
Lingulichnus verticalis
Hakes 1976	Spade to elongated-V shaped vertical burrow with the widest dimension at the top. Protrusive spreiten and a single projection down from the main chamber (interpreted as a pedicle trace).	Domichnia/Equilibrichnia	
Arenicolites statheri
Bather 1925	Simple, circular, vertical U-shaped burrow with parallel arms and no spreite. On bedding surfaces they appear as pairs of burrow apertures infilled with the overlying sediment.	Domichnia	
Catenichnus contentus
McCarthy 1979	Vertical, symmetrical, shallow U-shaped burrows with retrusive spreiten but without parallel arms. The burrows are enlarged at the aperture and oval shaped. Burrow is wider than deep.	Domichnia	
Diplocraterion parallelum
Torell 1870	Simple, circular, vertical U-shaped burrow with parallel arms and. spreiten. On bedding surfaces they appear as connected pairs of burrow apertures.	Domichnia/Equilibrichnia	
Rhizocorallium jenense
Zenker 1836	Straight, short U-shaped burrows with typically protrusive spreiten. Orientated oblique to the bedding plane.	Domichnia/Equilibrichnia	
Rhizocorallium irregulare
Mayer 1954	Long sinuous U-shaped burrows with protrusive spreiten. Mostly horizontal to bedding. On bedding surfaces they can also appear as pairs of burrow apertures both connected or disconnected.	Fodinichnia	
Teichichnus rectus
Seilacher 1955	Series of vertical spreiten. Spreiten have a curved base and are imperfectly stacked upon each other.	Fodinichnia	
Thalassinoides suevicus
(Rieth 1932)	Horizontal systems of irregular, subcylindrical, smooth burrows with dichotomous Y-shaped branching. Burrow diameter more or less consistent except for swellings at the site of branching.	Fodinichnia	
Planolites montanus
Richter 1937	Subcylindrical to cylindrical, unbranched, horizontal, sinuous burrows with a smooth surface and no internal structure.	Fodinichnia	
Palaeophycus tubularis
Hall 1847	Horizontal, thinly-lined subcylindrical to cylindrical, smooth burrows. Sediment fill is structureless.	Domichnia	
Helminthopsis cf. hieroglyphica Maillard 1887	Unbranched, loose, irregularly meandering, horizontal burrows, that commonly forms U-shapes. The burrow surface is smooth and unlined.	Fodinichnia/Pascichnia	
“Small stuffed burrows” Pollard 1981	Small spheroidal or subovate hypichnia. Occurs in clusters.	Domichnia/Cubichnia	
Chondrites intricatus (Brongniart 1823)	Small trace fossil composed of numerous downward radiating, straight branches. On both bedding and vertical surfaces the burrows appear as cluster of straight tubes and cross-sections of cylindrical tubes. Rarely, a main shaft from which the burrows radiate can be observed.	Fodinichnia/Agrichnia	
Taenidium isp. A	Thinly-lined, horizontal to oblique, cylindrical, sinuous tubes with regular meniscate backfill.	Fodinichnia	
Taenidium? isp. B	Thinly-lined, horizontal to oblique, cylindrical, sinuous tubes with regular meniscate backfill that also branch.	Fodinichnia	
Aulichnites parkerensis
Fenton and Fenton 1937	Sinuous to straight, unornamented bilobate trails with a narrow median furrow	Repichnia/Pascichnia	

Figure 6 Examples of the ichnogenera identified from the different bedding surfaces in the field.

P, Planolites, Ta, Taenidium, Te, Teichichnus, Rh, Rhizocorallium, Ca, Catenichnus, Sk, Skolithos, Ar, Arenicolites, Di, Diplocraterion, Th, Thalassinoides, Ch, Chondrites. Scale bar = 1 cm. Camera lens cap diameter = 5 cm.

Figure 7 Cluster analysis of the trace fossils from bedding planes from the Hiraiso Formation, Miyagi Prefecture, Japan.

Five different ichnological communities were qualitatively recognised based on the cluster analysis separated by dashed lines. The lithofacies each sample comes from is labelled. Facies Hd was subdivided into Hd (lower) and Hd (upper) based on log heights.

There were 863 burrows measured from bedding surfaces, with burrow diameters ranging between 1 to 26 mm, and the largest burrows belonging to Thalassinoides suevicus, and most of the burrow diameters being <10 mm (Fig. 8). The largest ichnospecies are Catenichnus contentus, Rhizocorallium irregulare, Skolithos linearis, and Thalassinoides suevicus. Temporal and spatial changes in the body sizes of ichnofossils will, therefore, be strongly affected by the presence/absence of certain ichnospecies. When the burrow sizes of ichnogenera are grouped, there is also a slight onshore-offshore gradient where the burrow sizes in lithofacies Hd and He are larger than the shallower lithofacies. This offshore size increase, however, may be related to the shift towards ichnocommunities dominated by Rhizocorallium irregulare and Catenichnus contentus as none of the ichnotaxa show significant increases in their median size (Fig. 9).

Figure 8 Jitter plot showing burrow diameters for each ichnospecies measured in the Hiraiso Formation.

The squares indicate the median size for each ichnospecies.

Figure 9 Jitter plot of burrow diameters of the most abundant ichnogenera across the different lithofacies.

The median diameters for each ichnospecies are shown as squares for each lithofacies.

Ichnofabric analysis

The polished slabs could broadly be grouped into ten different ichnofabrics (see Table 4; Fig. 10). Overall, these ichnofabrics show an onshore-offshore trend, where ichnofabrics 1 to 4 are dominantly found in the shallower lithofacies, and ichnofabrics 5–9 are dominantly found in the deeper lithofacies (Fig. 11). The range of values for ichnodiversity and the ichnofabric index are similar for all of the lithofacies, but lithofacies Hc, Hd, and He (offshore transition-distal offshore) are more likely to have higher ichnodiversity and ichnofabric values (Fig. 11).

Table 4 Descriptions of the different ichnofabrics recognised in this study.

Ichnofabric	Description	
1a. Unbioturbated, laminated ichnofabric	No bioturbation recorded and sediment laminations are distinct. Rare ostracods and crinoids are recorded. Some laminations record subtle bioturbation (II2).	
1b. Unbioturbated, massive ichnofabric	Structureless fine sandstone frequently with subcentimetre mud clasts forming thin horizons within the fabric. Occasionally, some rare indeterminate vertical traces and Chondrites.	
2. Vertical indeterminate traces ichnofabric	Dominantly bioturbated by vertical indeterminate traces. The vertical indeterminate traces look similar to the probable pedicle traces associated with lingulid brachiopods (see Zonneveld, Beatty & Pemberton, 2007). Other ichnofossils include: Arenicolites, Chondrites, Diplocraterion, Teichichnus, and Lingulichnus.	
3. Teichichnus ichnofabric	Alternations of courser beds that are either fine sand or randomly orientated shell material with silt to very fine sandstones that are highly bioturbated by Teichichnus. Less common traces include Rhizocorallium, Chondrites, Skolithos, vertical indeterminate fossils, and Planolites.	
4. Chondrites-Teichichnus ichnofabric	Alternating 1–3 cm beds of unbioturbated fine sand and silt to very fine sand. The finer sediments are dominantly bioturbated by Chondrites and occasionally extend into the underlying fine sand. Teichichnus cross-cuts the Chondrites burrows. All burrows are shallow tiering (1–3 cm depth).	
5. Rhizocorallium-Chondrites ichnofabric	Bioturbated but bedding distinct. Rhizocorallium and Chondrites alternate and are the dominant bioturbators. Rhizocorallium does not cross-cut other traces and occasionally Chondrites cross-cuts the Rhizocorallium traces.	
6. Rhizocorallium ichnofabric	Highly bioturbated fabric dominated by relatively large Rhizocorallium traces. Burrow depths exceed the sample size. The Rhizocorallium burrows cross-cut everything. Less common traces include Arenicolites, Chondrites, Teichichnus, and Planolites.	
7. Chondrites-Diplocraterion ichnofabric	Moderately bioturbated. Dominated by Chondrites and Diplocraterion, where Chondrites typically cross-cuts other traces. Less common traces include Rhizocorallium, Skolithos, and Planolites.	
8. Chondrites-Rhizocorallium ichnofabric	Highly bioturbated and dominated by Chondrites and to a lesser extent Rhizocorallium, where Rhizocorallium cross-cuts the Chondrites burrows. Less common traces include Lingulichnus, Planolites, and Teichichnus.	
9. Chondrites ichnofabric	Highly bioturbated fabric dominated by Chondrites, which also typically cross-cuts other ichnofossils. Less common traces include: Rhizocorallium, Diplocraterion, Planolites, and Teichichnus.	

Figure 10 Examples of the different ichnofabrics recognised in this study.

(A) Unbioturbated, laminated ichnofabric. (B) Unbioturbated, massive ichnofabric. (C) Vertical indeterminate traces (cf. Lingulichnus) ichnofabric. (D) Teichichnus ichnofabric. (E) Rhizocorallium-Chondrites ichnofabric. (F) Rhizocorallium ichnofabric. (G) Chondrites-Teichichnus ichnofabric. (H) Chondrites-Rhizocorallium ichnofabric. (I) Chondrites ichnofabric. Scale bar = 1 cm.

Figure 11 Ichnological data collected from the ichnofabric analysis between the different lithofacies.

(A) Ichnodiversity. (B) Ichnofabric Index (bioturbation). (C) Ichnofabric. (D) Burrow Depths. Note: burrow depths frequently extended beyond the sample size. These depths also do not account for sediment compaction.

Tiering depth also appears to show an onshore-offshore trend, where burrows that penetrate deeper into the sediment are more likely to be found in the deeper lithofacies (Fig. 11). The maximum burrow depths from the ichnofabrics is, however, affected by the thickness of the bed, i.e., in several instances the burrow depth exceeds the thickness of the bed and it cannot be completely measured, and in at least three samples exceeds 10 cm. In addition, based on the cross-sections of Rhizocorallium burrows, the burrow height is 34–40% of the burrow width, demonstrating that the sediment shows a high degree of compaction. Taking this compaction into account, the two deepest observed burrows, which was a Skolithos (10 cm) and Diplocraterion (7 cm), exceeded depths of 17.5 and 25 cm into the sediment, respectively. The compaction of the sediment would, however, have also increased the width of the burrow, so the maximum burrow depth is likely between 10–25 cm for Skolithos and 7–17.5 cm for Diplocraterion.

Shell bed analysis

Bedding planes that exposed body fossils were rare and when they do exist the fossils are present as moulds, which reduces the possibility of species identification. Eumorphotis iwanowi (Bittner 1899), Neoschizodus laevigatus (Goldfuss 1837), Entolium ussuricus (Bittner 1899), Bakevellia (Maizuria) cf. kamebei Nakazawa (1959), and cf. Entolium sp. were observed, but only two bedding planes exposed multiple specimens: 6.8 m at section 1a and 18.6 m at section 2. E. iwanowi is a relatively large Early Triassic bivalve species and in this study records a maximum geometric size of 56 mm and a median of 42 mm (Fig. 12). In addition to bivalves, moulds of the crinoid ossicles assigned to Holocrinus sp. are also preserved throughout the Hiraiso Formation (systematic taxonomy is given by Kashiyama & Oji (2004)). These crinoid ossicles are, however, particularly abundant at 65–70 m at section 3 and 105–110 m at section 4 in lithofacies Hc.

Figure 12 Histograms of the geometric median shell size of bivalve specimens from the Hiraiso Formation.

(A) Bedding plane at 6.8 m in section 1a. (B) Bedding plane at 18.6 m in section 2. Histogram bin width is 1 mm.

Abundant bivalve, gastropod, and crinoid fossils occur in carbonate rich lenses and gutter casts throughout the Hiraiso Formation, which preserves shell material. The mollusc shells typically appear as either black recrystallised shells, or the shell is replaced with pyrite framboids. The crinoid ossicles appear as white recrystallised calcite or are replaced with pyrite. The faunal composition could be studied using polished slab analysis but, because of the lack of reference material, specific identifications could not be determined. Fifteen different taxonomic groups were recognised with some of the groups representing multiple different undetermined species. The richness of samples ranges from 1 to 13 taxa. The cluster analysis combined with the SIMPROF test, shows that the polished slab samples based on their faunal composition can be grouped into 9 biofacies (Fig. 13). Of these 9 biofacies, biofacies 2–8 can be grouped into biofacies dominated by taxon-rich communities (richness: 11–13 taxa) and “Bivalve 2” (Fig. 13), which likely represent an infaunal genus (e.g., Neoschizodus or “Unionites”). The distribution of the different biofacies does not show any trends associated with lithofacies or log height (Fig. 13). This suggests that the sampled shell beds represent allochthonous communities, as the shell lenses are the result of storm events that redistribute the different taxa along the shelf smearing the original habitat partitions. In addition, the low taxonomic resolution of the polished slab analysis means that some of the species distributions observed in the field are not strictly recognisable. For example, Bivalve 3 is likely to be specimens of Eumorphotis iwanowi and is abundant in the shallower lithofacies, consistent with field observations for this species. Bivalve 3 is, however, also recognised from deeper lithofacies, which could be because this morphotype is reflected by multiple species with different lithofacies ranges.

Figure 13 Cluster analysis of the bioclasts from polished slab samples from the Hiraiso Formation, Miyagi Prefecture, Japan.

The cluster analysis together with the SIMPROF test identified 9 groups (labelled a–i) of samples that are statistically distinct. The different groups have been interpreted as different biofacies.

Geochemical proxies

Several redox sensitive metals (Mo, U, V) can be used as proxies for marine redox conditions (Tribovillard et al., 2006). Where, redox sensitive metals (Mo, U, V) are commonly more soluble under oxidising conditions, but get removed from the seawater column at the seawater-sedimentary interface under oxygen depleted-conditions, rendering them proxies for bottom water redox changes (Tribovillard et al., 2006). Mo and U concentrations of the Hiraiso Formation (Supplement S7) are consistently lower than average marine shale concentrations, relative to Post Archean Australian Shale (PAAS (Taylor & McLennan, 1985)). V concentrations are also generally below PAAS values, with the exception of two samples with slightly elevated V concentrations of up to 164 ppm. To account for potential lithologic affects, the redox sensitive elements were further normalised to Al. The Al-normalised values show very little variation for all three redox sensitive elements and no enrichment compared to PAAS, suggesting stable redox conditions.

V/Cr and Th/U ratios have been established as geochemical proxies for local marine redox conditions. Both V and Cr are redox sensitive elements and immobile under anoxic conditions, but the reduction of Cr occurs at a lower Eh boundary than the reduction of V. Based on this, V/Cr values <2 were defined as characteristic for oxic conditions, values from 2 to 4.25 as characteristic for dysoxic conditions and values >4.25 as characteristic for anoxic conditions (Jones & Manning, 1994). While Th is relatively unaffected by redox changes, U is immobilised under anoxic conditions resulting in U enrichment under reducing conditions as indicated by Th/U values <2 (Twitchett & Wignall, 1996). The Hiraiso Formation is characterised by V/Cr ratios between 2.71 and 4.25 and Th/U ratios between 1.94 and 3.59, suggesting that deposition occurred in a consistently (dys-)oxic environment (Fig. 14).

Figure 14 Cross plot of redox indices V/Cr and Th/U.

The ranges for V/Cr and Th/U are from Jones & Manning (1994) and Twitchett & Wignall (1996), respectively.

The Hg data only includes 12 samples that recorded Hg concentrations above the detection limit (>1 ppb) and no excursions in Hg concentrations were recorded.

Discussion

How old is the Hiraiso Formation?

The significance of the ecological state of the faunal communities identified from the Hiraiso Formation are dependent upon the age of the strata, i.e., if the Hiraiso Formation was deposited during the Smithian substage then the findings of an advanced ecological state will have implications for our understanding in the different pace of recovery between different regions, whereas if the strata are Spathian then that would suggest the recovery of marine ecosystems is ubiquitous between different locations and ocean basins. The age of the Hiraiso Formation was poorly constrained (see Geological Setting for details). A revision of the ammonoids and new material collected from the Hiraiso and Osawa formations now mean that the age of the Hiraiso Formation is more confidently interpreted than before. A recently sampled ammonoid specimen by Shigeta & Nakajima (2017) belonging to Tirolites indicated that the Hiraiso Formation corresponds to the base of the early Spathian. This agrees with the recent revision of the ammonoid material sampled from the base of the Osawa Formation, which is late early Spathian to early middle Spathian in age (Shigeta, 2021). Part of the assemblage is coeval with the Procolumbites Zone of western USA (Guex et al., 2010; Jenks et al., 2013) and South China (Galfetti et al., 2007; Ji et al., 2015). The newly sampled ammonoid specimens yield two proptychitid taxa among which Koninckitoides aff. posterius is identified (see Systematic Palaeontology). This genus is characteristic of the lower part of the early Spathian and for instance allows correlation with the early Spathian Tirolites-Amphistephanites Zone of the nearby basin of South Primorye (Zakharov & Smyshlyaeva, 2016). This interpretation is thus in agreement with the Tirolites occurrence reported by Shigeta & Nakajima (2017). In addition, positive excursions in Hg concentrations have been used as stratigraphical markers for the end-Permian mass extinction and “late Smithian thermal Maximum” (Grasby et al., 2016), but positive excursions in Hg concentrations were not recognised in this study. Taken together, these data confirm an early Spathian age for the Hiraiso Formation.

Ecological state of the Hiraiso Formation

A previous assessment of the ecological state of the Hiraiso Formation fossil communities led Kashiyama & Oji (2004) to conclude that the Hiraiso Formation does not record an advanced state of ecological recovery following the end-Permian mass extinction. This was because they only recovered a low-diversity of benthic taxa (10 taxa). Our study of the shelly faunal communities does not negate the finding that the Hiraiso Formation preserves a low-diversity shelly fauna, but we do not, however, follow the conclusion that it represents a lack of ecological recovery following the end-Permian mass extinction. Instead, the low richness of the Hiraiso Formation is interpreted as a consequence of poor preservation and the rare exposure of bedding planes with fossil shells. Support for poor preservation conditions for skeletal material comes from the presence of ichnofossils with a known producer and the absence of that producer, i.e., lingulids have not been reported from the Hiraiso Formation despite the high abundance of Lingulichnus ichnofossils. The Hiraiso Formation is also exclusively composed of siliciclastic material, which is known to hamper the preservation of aragonitic and calcitic material due to the undersaturation of pore waters with respect to carbonate (Aller, 1982). In addition, the bioturbated and porous nature of the Hiraiso Formation means that alkalinity is prevented from being built up during sulphate reduction and makes the sediment-water interface an environment that is highly corrosive to skeletal material (Aller, 1982; Best, 2008). This is supported by the occurrence of shell lenses, which are associated with micrite and the replacement of skeletal material with pyrite.

Even though species richness is interpreted to be an unreliable measure of ecological recovery for the Hiraiso Formation, other ecological attributes such as the presence of key taxa, body size, tiering and ichnological record can provide insights into the ecological state of shallow marine ecosystems (Twitchett, 2006). The presence of key species and large body sizes have also been highlighted before for the Hiraiso Formation by Shigeta & Nakajima (2017) (from a stratigraphical point-of-view) who noted the presence of Holocrinus and the relatively large bivalve Eumorphotis iwanowi indicated an ecological state similar to Spathian fossil assemblages from Primorye, Russia. The bivalve body sizes and burrow diameters recorded in this study are typical for the Early Triassic meaning that they do not indicate “full” ecosystem recovery: some burrows of Thalassinoides exceed 20 mm but otherwise most of the recorded burrows are <10 mm (Fig. 8) consistent with other Early Triassic ichnofaunas (Twitchett, 1999, 2006; Beatty, Zonneveld & Henderson, 2008; Zonneveld, Gingras & Beatty, 2010; Hofmann et al., 2011; Foster et al., 2015, 2018b). Likewise, the bivalve Eumorphotis iwanowi from the Hiraiso Formation is relatively large for Early Triassic bivalves, but it is within the range of large Claraia species recorded from the Griesbachian and Dienerian (Twitchett, 2007; Foster, 2015; Foster et al., 2020) and large Eumorphotis species recorded from the Olenekian (Broglio-Loriga et al., 1990; Hautmann et al., 2013; Kolar-Jurkovsek et al., 2013; Foster, 2015).

Index taxa that have been classified as representing an advanced ecological state for the Early Triassic that have been identified from the Hiraiso Formation includes, Holocrinus, Rhizocorallium, and Thalassinoides. The ichnogenera Rhizocorallium and Thalassinoides are interpreted as index taxa because they are thought to have been produced by crustaceans, which are some of the last animal groups to reappear after hypoxic events (Twitchett & Barras, 2004). Luo et al. (2020) also highlight that Rhizocroallium and Thalassinoides have multiple tracemakers and that these ichnogenera can only be used as bioindicators of an advanced ecological state if they were produced by malacostracan crustaceans. Scratch marks were not recorded with Rhizocorallium burrows in this study and either crustaceans or worm-like organisms could have produced the burrows (Rodríguez-Tovar et al., 2012). Swellings at the branches for Thalassinoides suevicus are suggested to indicate a malacostracan crustacean tracemaker (Luo et al., 2020), suggesting that the presence of Thalassinoides suevicus together with Holocrinus in the Hiraiso Formation indicates an advanced stage of ecological recovery. These are genera and ichnogenera that are typical of Spathian successions (Twitchett & Wignall, 1996; Twitchett, 1999; Hofmann et al., 2013; Foster et al., 2015, 2017, 2018a), but have also been recorded from a few late Griesbachian (Twitchett et al., 2004; Hofmann et al., 2011; Foster et al., 2017; Brosse et al., 2019) and Smithian successions (Oji, 2009). In addition to these index taxa, there are also some “complex trace fossils” from the Hiraiso Formation which include forms that meander (Helminthopsis cf. hieroglyphica, Rhizocorallium irregulare), produce both regular branching and radiating spreitin (Chondrites intricatus), and both regular backfill and branching (Taenidium isp. 2). Early Triassic ichnofossil databases also suggest that Chondrites is an Early Triassic Lazarus ichnogenus and that the Thalassinoides burrow sizes are the largest recorded for the Early Triassic (Luo et al., 2020; Cribb & Bottjer, 2020). The overlying Osawa Formation also records the presence of marine reptiles (Tokio & Masafumi, 1978), which shows that marine predators that are often indicative of a fully functioning marine ecosystem (Chen & Benton, 2012), were also present on the South Kitakami terrane during the Spathian.

Tiering above and below the sediment has also been suggested as a marker of ecological recovery (Twitchett, 1999), because tiering was reduced to within a few centimetres of the sediment-water interface after the mass extinction (Bottjer & Ausich, 1986). The erect tier is represented by the crinoid Holocrinus, but the ossicles are relatively small and are assumed to not occupy the full range of available space above the sediment-water interface as recorded in the Anisian (Hagdorn & Velledits, 2006; Foster et al., 2015). With few exceptions, tiering below the sediment is typically within 12 cm of the sediment-water interface during the Early Triassic (Luo et al., 2020), but previous studies do not tend to take compaction of the sediment into account. In addition, some exceptionally deep infaunal traces recorded from the Early Triassic (e.g., Arenicolites from Svalbard (Wignall, Morante & Newton, 1998)) appear to be multiple burrows overlapping from multiple beds rather than being a single burrow. In the Hiraiso Formation most burrows are within 10 cm of the bedding surface, and when compaction is accounted for, maximum burrow depths are as deep as 25 cm. This suggests that the deep tier is occupied in the Hiraiso Formation, but infaunal organisms are not penetrating as deep into the sediment as recorded in Middle Triassic successions (Luo et al., 2020).

Taken together the presence of key taxa, complex ichnofossils, relatively large body and burrow sizes, and both deep infaunal and erect tiers in the Hiraiso Formation suggests an advanced ecological state following the end-Permian mass extinction. This ecological state is comparable to stage 3 of Twitchett (2006) recovery model that has been recorded in late Griesbachian and Spathian faunal communities elsewhere, but also records unique elements. This also suggests that the early Spathian represents a globally synchronous interval of recovery following the end-Permian mass extinction. An important aspect of this advanced ecological state is how quick it appears after the deleterious environmental conditions associated with the “late Smithian Thermal Maximum” ended. In addition, based on how long it took for the first ecological complex faunal ecosystems to appear after the end-Permian mass extinction, early Spathian recovery from the “late Smithian Thermal Maximum” is more advanced than would be expected. The differences in the pace of recovery between the two events is possibly related to the duration of the unfavourable environmental conditions, where the “late Smithian Thermal Maximum” was a shorter event. Alternatively, even though the late Smithian marks an interval of biotic crisis for conodonts and ammonoids (Brayard et al., 2006; Stanley, 2009; Wu et al., 2019) and a significant turnover in benthic communities (Hofmann et al., 2014; Hofmann, Hautmann & Bucher, 2015; Foster et al., 2015, 2017, 2018a), it may have not been a biotic crisis for the benthos.

Implications for the habitable zone hypothesis

An investigation on the distribution of diverse ichnofaunal communities led Beatty, Zonneveld & Henderson (2008) and Zonneveld, Gingras & Beatty (2010) to hypothesise that wave-aerated settings on shallow marine shelves created oxygenated habitats that could support a diverse and large macrofauna during periods when oxygen minimum zones expanded into shallow settings, dubbed the habitable zone hypothesis. Support for a habitable zone during the Early Triassic also comes from tropical and subtropical settings in Nunavat, Canada (Beatty, Zonneveld & Henderson, 2008; Proemse et al., 2013), western Canada (Beatty, Zonneveld & Henderson, 2008), western USA (Fraiser & Bottjer, 2009; Pietsch, Mata & Bottjer, 2014; Woods et al., 2019), Lombardy, Italy (Foster et al., 2018a), Dolomites, Italy (Foster et al., 2017; Posenato, 2019), South China (Shi et al., 2015; Feng et al., 2017), Svalbard (Foster, Danise & Twitchett, 2017), and Australia (Chen, Fraiser & Bolton, 2012; Feng et al., 2021). Even though, many successions support this hypothesis, the habitable zone or an oxygenated setting does not guarantee rapid recovery (Twitchett et al., 2004; Foster et al., 2015; Feng et al., 2017).

The Hiraiso Formation and the overlying Osawa Formation allow us to test the presence of a habitable zone during the early Spathian on the South Kitakami terrane. The Hiraiso Formation provides a transect of a shallow marine shelf with deposition from the lower shoreface to distal offshore settings and the overlying Osawa Formation represents a basinal setting. Our data on the distribution of ichnofossils from the Hiraiso Formation shows a number of characteristics that are consistent with the habitable zone hypothesis: (a) a high-diversity of trace fossils and high degree of bioturbation is recorded, (b) ichnofaunal communities show an onshore-offshore trend with the dominance of different ichnospecies (Fig. 7), (c) ichnofaunal communities include ichnospecies with relatively large burrow diameters (Fig. 8), and (d) relatively large burrow depths, i.e., significant infaunal tiering during the Early Triassic (Fig. 11).

In addition, the interpretation that the body fossils also indicate a relatively advanced ecological state suggest that the Hiraiso Formation enabled the recovery of marine ecosystems in shallow, oxygenated settings. When comparing the V/Cr and Th/U ratios recorded for the Hiraiso Formation they overwhelmingly fall in the range of dysoxic (weakly oxidised) for V/Cr and oxic for Th/U (Fig. 14), supporting sediment deposition under upper dysoxic to oxic conditions. For the habitable zone hypothesis to be accepted it also has to be shown that marine settings below the action of waves were uninhabitable for benthic communities. Ichnological studies of the Osawa Formation show that it records both intensely bioturbated and unbioturbated intervals in a basinal environment (Yamanaka & Yoshida, 2007; Yoshizawa et al., 2021). It is important to note that these intensely bioturbated intervals are restricted to thin laminae. In addition, investigations of redox proxies suggest that these highly bioturbated intervals correspond with oxygenated intervals, whereas lower dysoxic intervals correspond with unbioturbated sediment during the Spathian (Yamanaka & Yoshida, 2007; Yoshizawa et al., 2021). This suggests that the South Kitakami terrane experienced intervals of a shallow oxygen minimum zone that restricted benthic faunas to shallow marine settings throughout the Spathian.

A comparison of the ichnofauna and bioturbation between the Osawa and Hiraiso formations also shows some clear differences. In the Osawa Formation only three microscopic ichnogenera are recorded Chondrites (diameter 0.2 mm), Planolites (diameter 1–5 mm), and Phycosiphon (diameter 0.1–0.6 mm) (Yamanaka & Yoshida, 2007). Whereas in the Hiraiso Formation 15 ichnogenera were identified with considerably larger diameters (Fig. 8). The presence of Chondrites and Planolites in the basinal setting are also consistent with our finding in the Hiraiso Formation that these ichnogenera are more abundant in the deeper lithofacies. Furthermore, even though the lower Osawa Formation was “oxic”, the bottom waters were not sufficiently oxygenated to sustain the metabolic activities of a diverse and large macrofauna that was present in the shallower facies of the Hiraiso Formation. The controls on the degree of bioturbation also appear to be different between the two formations and depositional settings. In the Hiraiso Formation the degree of bioturbation is controlled by the sediment flux onto the shelf, which is shown by the shallower, more wave-agitated lithofacies recording lower levels of bioturbation (Fig. 11). Whereas, in the distal offshore to basinal setting of the Hiraiso and Osawa formations the degree of bioturbation is controlled by oxygen availability, i.e., bioturbation is restricted to low oxic intervals. This onshore-offshore gradient of oxygen availability and distribution of bioturbation is consistent with the habitable zone hypothesis. The Hiraiso and Osawa formations were also deposited during the early Spathian, which was an interval of climate cooling and it is inferred that oxygen minimum zones retreated during this interval (Song et al., 2019). This suggests that despite the cooler early Spathian climate, oxygen minimum zones were still shallow during the early Spathian.

A problem with the habitable zone hypothesis is the outstanding question of: can it be considered a refuge for benthic marine ecosystems unique to anoxic events? Dissolved oxygen profile gradients from the Black Sea (restricted subtropical basin), Gulf of Mexico (semi-enclosed subtropical seaway), and Arabian Sea (open tropical ocean) show that low oxic conditions begin at 25–50 m water depths in these settings (Dashtgard & MacEachern, 2016). In addition, these low oxic conditions in modern settings are sufficient to support benthic microfauna, but insufficient to sustain a macrofaunal community (Dashtgard & MacEachern, 2016). Therefore, given the tropical-subtropical depositional setting of the South Kitakami terrane, the observed onshore-offshore gradient in infaunal diversity and body size is consistent with observations for both modern analogues and ancient analogues (see also the Middle Triassic Sunset Prairie Formation (Furlong et al., 2018)). The significance of unbioturbated mudstones and presence of oxygen stress below wave base in Early Triassic subtropical and tropical basins may, therefore, have been overstated and not unique to anoxic events.

Delayed recovery in oxygenated settings

The Hiraiso Formation records deposition in a weakly oxygenated shallow marine setting that enabled the recovery of benthic marine communities to an advanced ecological state. This benthic fauna does not, however, record full recovery and does not record the same level of recovery as some Middle Triassic successions (Zonneveld, Gingras & Beatty, 2010; Foster & Sebe, 2017; Friesenbichler et al., 2018, 2021). While intermittent drops in bottom water oxygenation to lower dysoxia could have hindered the recovery of benthic communities, other environmental factors likely influenced the pace of recovery during the Early Triassic as well. A delayed recovery within the proposed habitable zone has also been reported by previous studies (Twitchett et al., 2004; Fraiser & Bottjer, 2009; Foster et al., 2015; Feng et al., 2017; Foster et al., 2018b), and these authors suggested that other environmental factors, such as elevated carbon dioxide, high sediment flux and thermal stress, must have contributed to the delayed recovery. In the absence of independent proxies for environmental stress, high thermal stress is commonly cited as the most parsimonious explanation as a critical environmental stressor (Sun et al., 2021). Oxygen isotope data from conodont apatite suggests that the entire Early Triassic was hotter than the Changhsingian and there were thermal maximums in the mid-Griesbachian and late Smithian (Sun et al., 2012; Romano et al., 2012). This led Song et al. (2014) to propose that, based on the temperature gradient of the Black Sea, thermal stress inhibited organisms from the shallowest settings above wave base. Temperature gradients from the modern, open, tropical ocean setting of the Arabian Sea and the subtropical, restricted setting of the Black Sea show that temperature only decreases by 6–8 °C over 200 m water depth (Torres, Grigsby & Clarke, 2012). It is, therefore, extremely unlikely that only the shallowest part of the habitable zone was thermally stressed and that thermal stress within the whole habitable zone likely limited the metabolic activities of macrofauna during the Early Triassic. The impact of thermal stress means that, once an organism exceeds its pejus temperature, its physiological performance will decline and will result in the worsening of oxygen supply to the organism (Verberk et al., 2016). If you combine the effects of thermal stress on the aerobic scope of marine organisms, and consider that the depositional environment of the Hiraiso Formation was low oxic then the combined effects of limited oxygen availability and thermal stress could explain why the Hiraiso Formation assemblages typically record small body sizes.

An alternative explanation for a delayed recovery comes from theoretical diversification models. Theoretical models suggest that niches emptied by mass extinctions should refill rapidly after extinction stresses ameliorate (Erwin, 2001; Hautmann, 2014). This reoccupation is initially exponential but slows as environmental carrying capacity is approached (Erwin, 2001; Hautmann, 2014). This exponential increase in diversity is, however, delayed if the magnitude of species of loss is beyond a threshold that inhibits diversification processes, such as intraspecific competition (Hautmann et al., 2015). It has also been shown that diversity partitioning and a hyperbolic increase in species diversity of benthic invertebrates is delayed until the Middle Triassic (Hofmann et al., 2014; Friesenbichler, Hautmann & Bucher, 2021). The delayed recovery signal for benthic organisms during the Early Triassic could, therefore, be in part due to the magnitude of species loss and the suppression of biotic interactions that drives the diversification of both new and larger species. It should be noted, however, that the Early Triassic fossil record is notoriously poor (Fraiser, Clapham & Bottjer, 2010), including the fossil preservation of the Hiraiso Formation. Discoveries such as the silicified fauna from Lusitaniadalen (Svalbard) (Foster, Danise & Twitchett, 2017), semi-silicifed faunas from Oman (Twitchett et al., 2004; Brosse et al., 2019) and the Paris Biota (USA) (Brayard et al., 2017) demonstrate how the biases of the fossil record affect perceptions of the timing and patterns of diversity of life through the Early Triassic. These new collections are revealing a greater diversity in Early Triassic oceans, including several Lazarus taxa, and an earlier onset of the post-Permian radiation than previously thought (Foster, Danise & Twitchett, 2017; Brayard et al., 2017).

Conclusion

Palaeoecological analysis of the Hiraiso Formation shows that the South Kitakami terrane housed a benthic fauna that represents an advanced ecological state in the early Spathian. This fauna shows that the early Spathian records a globally synchronous stage of recovery for benthic ecosystems. The ichnocommunity preserves an onshore-offshore gradient with clear bathymetric ranges for the different ichnospecies identified from the bedding surfaces, increased bioturbation, deeper infaunal tiering and larger burrow sizes in the offshore to distal offshore setting. The body fossils do not, however, preserve distinct bathymetric ranges which is interpreted to be a consequence of the shell lenses representing allochthonous assemblages. The restriction of a relatively large and diverse benthos to shallow settings is consistent with the habitable zone hypothesis and the distribution of macrofauna in modern tropical and subtropical settings. This also suggests that the shallow marine habitable zone is not restricted to anoxic events. This ecological data also suggests that something other than oxygen stress delayed recovery in shallow marine environments, which could be temperature stress or the amount of time to recover from a high magnitude mass extinction event.

Appendix

Figure A1 Measured sections of the Hiraiso Formation along the Hiraiso Coastline, Miyagi Prefecture, Japan.

Ichnofabric Index (II) after Droser & Bottjer (1986). Grain size scale: C, clay; S, siltstone; VF, very fine sand; F, fine sand. Colour in the lithology column refers to the rock colour observed. Seds, sedimentary structures. The GPS coordinates for the base of each log are also shown.

Figure A2 Measured sections of the Hiraiso Formation along the Hiraiso Coastline, Miyagi Prefecture, Japan.

Ichnofabric Index (II) after Droser & Bottjer (1986). Grain size scale: C, clay; S, siltstone; VF, very fine sand; F, fine sand. Colour in the lithology column refers to the rock colour observed. Seds, sedimentary structures. For the logged sections 2–5 the composite log height is given. The GPS coordinates for the base of each log are also shown.

Figure A3 Measured sections of the Hiraiso Formation along the Hiraiso Coastline and Kamiwarizaki, Miyagi Prefecture, Japan.

Ichnofabric Index (II) after Droser & Bottjer (1986). Grain size scale: C, clay; S, siltstone; VF, very fine sand; F, fine sand. Colour in the lithology columnrefers to the rock colour observed. Seds, sedimentary structures. For the logged sections 2–5 the composite log height is given. The GPS coordinates for the base of each log are also shown.

Figure A4 Measured sections of the Hiraiso Formation along the Hiraiso Coastline and Kamiwarizaki, Miyagi Prefecture, Japan.

Ichnofabric Index (II) after Droser & Bottjer (1986). Grainsize scale: C, clay; S, siltstone; VF, very fine sand; F, fine sand. Colour in the lithology column refers to the rock colour observed. Seds = sedimentary structures. For the logged sections 2–5 the composite log height is given. The GPS coordinates for the base of each log are also shown.

Supplemental Information

Supplemental Information 1 Ichnofossil relative abundances.

Matrix containing the relative abundances of the different ichnofossils recorded from the bedding surfaces.

Click here for additional data file.

Supplemental Information 2 Ichnofossil burrow measurements.

Dataset containing the burrow measurements of the different ichnofossils recorded from the bedding surfaces.

Click here for additional data file.

Supplemental Information 3 Ichnofabric data.

Dataset containing the ichnofabric data of the different ichnofabrics recorded from the polished slabs.

Click here for additional data file.

Supplemental Information 4 Bivalve size data.

Dataset containing the bivalve body size data from the bedding surfaces.

Click here for additional data file.

Supplemental Information 5 Skeletal relative abundances.

Matrix containing the relative abundances of the different skeletal fossils recorded from the polished slabs.

Click here for additional data file.

Supplemental Information 6 Geochemical data from the Hiraiso Formation.

Dataset containing the elemental data.

Click here for additional data file.

Supplemental Information 7 Redox proxy data used in this study.

Dataset containing the Mo/Al (ppm/wt%) U/Al (ppm/wt%) V/Al (ppm/wt%) Th/U and V/Cr ratios.

Click here for additional data file.

William Foster would like to thank: Mao Kato, Hiroki Kawai, Yuki Wakasugi, Prof. Shin-Ichi Fujiwara, and Prof. Sachiko Nishida for their extremely generous hospitality whilst conducting this research in Japan; the Japanese Society for the Promotion of Science (JSPS) for their cultural integration programme and the opportunity to meet his majesty the Emperor Akihito and her majesty the Empress Michiko of Japan; the Kobayashi Family who volunteered to provide lodgings during the cultural integration programme. Amanda Godbold would like to thank Prof. Yukio Isozaki for being an incredible host and mentor during her stay in Japan. She would also like to thank Sena Kono, and Dr. Hikaru Sawada for their generous hospitality. Finally, Amanda would like to express her gratitude to the JSPS for creating an enriching experience and to the Fujita family for providing lodging during the cultural integration programme. We would like to thank Yasunari Shigeta, Michael Hautmann and an annomyous reviewer for their constructive comments that improved the original version of this manuscript. We would also like to thank Yasunari Shigeta for accessioning the ammonoid specimens. We would also like to thank the National Park for the sampling permissions required for this study.

Additional Information and Declarations

Competing Interests

Author Contributions

Field Study Permissions

Data Availability

The authors declare that they have no competing interests.

William J. Foster conceived and designed the experiments, performed the experiments, analyzed the data, prepared figures and/or tables, authored or reviewed drafts of the article, and approved the final draft.

Amanda Godbold performed the experiments, authored or reviewed drafts of the article, and approved the final draft.

Arnaud Brayard performed the experiments, authored or reviewed drafts of the article, and approved the final draft.

Anja B. Frank performed the experiments, analyzed the data, prepared figures and/or tables, authored or reviewed drafts of the article, and approved the final draft.

Stephen E. Grasby performed the experiments, authored or reviewed drafts of the article, and approved the final draft.

Richard J. Twitchett performed the experiments, authored or reviewed drafts of the article, and approved the final draft.

Tatsuo Oji conceived and designed the experiments, performed the experiments, authored or reviewed drafts of the article, and approved the final draft.

The following information was supplied relating to field study approvals (i.e., approving body and any reference numbers):

Field work was approved by the Ministry of the Environment, Tohoku Regional Environment Office and logistics organised by Nagoya University.

The following information was supplied regarding data availability:

The raw data is available in the Supplemental Files.

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
