# Peer review of "Palaeoecology of the Hiraiso Formation (Miyagi Prefecture, Japan) and implications for the recovery following the end-Permian mass extinction"

_PeerJ, doi:10.7717/peerj.14357_

## Round 0.1 · original submission · Minor Revisions

You provide a comprehensive and thorough study of the Hiraiso Formation and its importance for better understanding the recovery of the EPME. I would love to see this work published but there are some crucial points which need to be addressed before publication. The main points are:

Sampling permissions: You need to clearly state if the necessary permissions were obtained to sample this area and clearly mention/describe those in the Material and Methods section (compare reviewer 1)

Ammonoid assignment and illustration: Fig. 5A of specimen NMNS PM35937 needs to be sharp/in focus throughout so that the suture line is visible (compare reviewer 3). Reviewer 1 also raised issues in assigning this specimen to Koninckitoides posterius. Please clarify these points.

Intrinsic factors shaping the pace of recovery: I agree with reviewer 2 that a clearer distinction should be made between local abiotic environmental conditions and the state of recovery. A related point is the difference between environmental/ecological processes and evolutionary processes like recovery (see reviewers 2, 3). Reviewer 2 kindly suggested literature which might be relevant in this context.

Habitable Zone hypothesis: I agree with reviewer 3 that it is not clear why the patterns you observed should be specific to the Early Triassic habitable zone as opposed to those expected in normal onshore-offshore setting during other times. For this part to be more convincing, you also need to include discussion of SST/thermal stress (compare 3).

Osawa Formation: Are you certain about the assignment to the Hiraiso Formation above 221 m in Supplementary Figure 18? It is likely already part of the Osawa Formation (compare reviewer 1).

Typographical/Formatting issues: please make sure Koninckitoides is correctly spelled (compare reviewer 1)

Figures: In addition to the sharpness/focus issue of Fig. 5A mentioned before, please also make sure the delineation for ‘lower shoreface’, ‘offshore transition’, and ‘distal offshore’ at the top of the figure line up with the definition of the zones based on FWWB and SWB (compare reviewer 3)

Please make sure these and all other points raised by the reviewers are considered and addressed. I feel most of these points if not all are straightforward to address.

I look forward to obtaining the revised manuscript.

·

Basic reporting

I rated this manuscript as a quality work, but it should be much highly rated in potential if the manuscript provided much more detailed discussion of ammonoid systematics.

Experimental design

The study area was designated as Quasi-national park on March 30, 1979, and later designated as National park from March 31, 2015. Therefore, in order to collect rocks including fossils from this area, it is necessary to obtain permission from the Ministry of the Environment in Japan. In addition, it is obligatory to submit a report after sampling. If the rock samples collected for this study were obtained through formal procedures, it should be clearly stated in “Material and methods” or other chapters. If the rock samples were collected without permission, it should not be used for research.

Validity of the findings

line 255: Konincktoides is a misspelling. The correct spelling is Koninckitoides.

line 255: Koninckitoides was named by Dagys and Ermakova (1988), not Popov (1961)

line 293: 1969, not 1959.

line 290: add text-fig. 9d.

line 291: add text-fig. 52.

line 292: add text-fig. 53.

line 295: add text-figs. 2, 3a.

line 302: It is useful for the reader to describe the features of NMNS PM35937 in “Description”.

Line 305 to 317: You should discuss why Koninckites dolosus is a synonym for Koninckites posterius in “Remarks”.

line 288 to line3 17: The assignment of the specimen NMNS PM35937 to Koninckitoides posterius is questionable. The first lateral lobe of NMNS PM35937 is deep and narrow, but those of the holotype and some illustrated specimens of Koninckitoides posterius are shallower and wider than NMNS PM35937 (see Popov, 1961, text-fig. 9d; Ermakova, 1981, text-fig. 52; Dagys and Ermakova, 1988, text-figs. 2, 3a). The first and second lateral saddles of NMNS PM35937 are more elongated than the holotype and some illustrated specimens of Koninckitoides posterius. The characteristics of the suture line of NMNS PM35937 are same as those of the suture lines of Pseudaspidites popovi and Koninckitoides solus. The umbilical shoulder of NMNS PM35937 seems to be angular, which has the same characteristics as Pseudaspidites popovi and Koninckitoides solus, but the umbilical shoulder is round in Koninckitoides posterius. Shell surface ornamentation on NMNS PM35937 is similar to that of Pseudaspidites popovi in having radial folds, but differs from Koninckitoides solus which has dense, weakly sinuous ribs. However, they differ in shell diameter and it is necessary to understand the change in ornamentation through growth. As mentioned above, the NMNS PM35937 looks similar to the Pseudaspidites popovi in terms of sutures, morphology and ornamentation, but it is also necessary to compare Koninckitoides solus with the same size. Because there is also a problem with generic assignment of Pseudaspidites popovi, NMNS PM35937 may be better to treat as “Proptychitid gen. et. sp. indet.

Figure 18: The part above 221 meters should be the Osawa Formation, because the Osawa Formation is mainly composed of siltstones and mudstones and lacks macroscopic bioturbation and benthic invertebrate fossils.

·

Basic reporting

This paper provides new information on the distribution of rich trace fossil communities and associated shell beds along a shelf gradient in the late Early Triassic, which was a crucial time interval in the recovery from the greatest Phanerozoic mass extinction. These data have not been available for the Hiraiso Formation before and are very useful for a better understanding of the environmental conditions at that time.

Experimental design

The documentation is very thorough throughout. The article meets 'aims and scope' of the journal. The structure of the ms is clear. However, I miss evidence for the postulated high seawater temperature (Iines 596 - 617) that stems from the studied site , see 'additional comments' below.

Validity of the findings

The discussion ignores the role of intrinsic biotic factors in shaping the pace of recovery. Please see my detailed comments below.

Additional comments

My main criticism refers to the discussion of the recovery state, which should make a clearer distinction between local abiotic conditions and the state of recovery, which is an evolutionary process. More precisely, a central argument in the discussion of the recovery state is the presence of Holocrinus, Rhizocorallium, and Thalassinoides, which are seen as ‘index taxa for advanced recovery’ (line 479). Yet, these taxa did not evolve in the Spathian, but have been present since the Griesbachian (line 492). I agree with you that they might be indicative for the absence of environmental stress, but since they already existed before (in older strata), their presence or absence simply reflects local environmental conditions, not the state of the evolutionary process of recovery. Consider that your method would give the same results if applied e.g. to oxic versus anoxic basins in the Late Jurassic. In the discussion of the ‘habitable zone hypothesis’ (lines 584-595), you make exactly this point, but unfortunately, you do not extend it into the rest of the discussion.

I acknowledge that the chapter is correctly labelled ‘Ecological state of the Hiraiso Formation’, but from line 479 onward, the ‘ecological state’ is equated with ‘recovery state’. Please make consistently clear that your data primarily indicate the local ecological state.

Related to this point is a biased discussion on ‘delayed recovery in oxygenated settings’. The authors discuss the observation that benthic faunas were not fully recovered even where oxygen was not a limiting factor and suggest alternative kinds of environmental stress (primarily extremely hot temperatures) as an explanation. However, this explanation ignores the intrinsic dynamic of evolutionary processes. Life cannot respond by immediate diversification after relaxation of environmental stress because rates of speciation are primarily controlled by the intensity of biotic interactions (chiefly interspecific competition), which was reduced as a consequence of the extinction event (Hautmann et al. 2015 and refs therein; Friesenbichler et al. 2021). A delay in the recovery even after a return to ‘normal’ environmental conditions is a prediction from basic ecological and evolutionary theory. It is still possible that there were detrimental conditions in the early Spathian and that the ‘biotic’ delay occurred later and added to the overall duration of the delay, but this interpretation would require substantiation by temperature proxies from your samples. At least, you should state that there are two alternative explanations for your observation that need to be tested.

A number of additional minor remarks:

lines 43 – 51:
It should be added that plants did not suffer a global mass extinction at the end of the Permian (Nowak et al. 2019). Although I agree that a majority agrees about rapid global warming, a short global ice age has recently also been suggested (Baresel et al. 2017), which should be mentioned.

lines 57 – 58:
The left branch of the diversification curve that corresponds to the ‘delayed’ recovery is an integral part of the hyperbolic function. See my comment on ‘delayed recovery’ from above. You could rephrase: “…, whereas bivalves and snails show a hyperbolic rediversification pattern with an Early Triassic delay and an explosive radiation in the Middle Triassic.”

line 61: Please cite here also Senowbari et al. (1993), who were the first who documented and discussed the reappearance of reefs in detail.

line 63:
You switch here from ‘ecological attributes’ to the conclusion of ‘diverse’ ecosystems. Diversity is usually understood as richness, and you cannot conclude from tiering, body size etc. on richness. Please clarify.

lines 99, 529:
For the western US, Hofmann et al. (2014) report “some evidence for short-term environmental disturbances during the Smithian–Spathian transition”, but no taxonomic turnover.

line 366:
‘size of the sample’ is a bit misleading in this context. I think you refer to the thickness of the beds, right?

line 378:
I know that poorly preserved pectionoids are very difficult to identify and I also acknowledge that you use open nomenclature, but “Eopecten” is a highly unlikely candidate. Is there no alternative option among the common Early Triassic pectinoid genera?

line 404:
I wonder how you can distinguish whether these geochemical proxies measure redox conditions in the water column or in the sediment? This is a significant difference for the interpretation. Could you please specify this point?

line 478:
The largest Spathian Eumorphotis is (to my knowledge) Eumorphotis cf. gronensis described by Kolar-Jurkovsek et al. (2013, fig. 7). This beast is larger than 8 cm in height.

line 573:
“bottom waters were not sufficiently oxygenated to sustain the metabolic activities of a diverse and large macrofauna”. (1) Was a diverse and large macrofauna already evolved in the early Spathian? You cannot sustain something that has gone extinct before!
(2) Please see comment on line 404.

line 595: YES!!

line 621:
Maybe I missed it, but where in the ms have you demonstrated that “the early Spathian records a globally synchronous stage of recovery for benthic faunas”? I think that it it hard to demonstrate that the trace fossil communities correspond to the same recovery state than shelly faunas elsewhere.

I hope that you find these comments constructive.

Best wishes,

Michael Hautmann

Refs:

Baresel, B., Bucher, H., Bagherpour, B., Brosse, M., Guodun, K., & Schaltegger, U. (2017). Timing of global regression and microbial bloom linked with the Permian-Triassic boundary mass extinction: implications for driving mechanisms. Scientific Reports, 7(1), 1-8.

Friesenbichler, E., Hautmann, M. & Bucher, H. (2021). The main stage of recovery after the end-Permian mass extinction: taxonomic rediversification and ecologic reorganization of marine level-bottom communities during the Middle Triassic. PeerJ 9:e11654

Hautmann, M., Bagherpour, B., Brosse, M., Frisk, Å., Hofmann, R., Baud, A., Nützel, A., Goudemand, N. & Bucher, H. (2015). Competition in slow motion: The unusual case of benthic marine communities in the wake of the end-Permian mass extinction. Palaeontology 58(5), 871–901.

Kolar-Jurkovsek, T., Vuks, V. J., Aljinovic, D., Hautmann, M., Kaim, A., & Jurkovsek, B. (2013). Olenekian (Early Triassic) fossil assemblage from eastern Julian Alps (Slovenia). Annales Societatis Geologorum Poloniae 83(3), pp. 213–227.

Nowak, H., Schneebeli-Hermann, E., & Kustatscher, E. (2019). No mass extinction for land plants at the Permian–Triassic transition. Nature communications, 10(1), 1-8.

Senowbari-Daryan B, Zühlke R, Bechstädt T, Flügel E. (1993). Anisian (Middle Triassic) Buildups of the Northern Dolomites (Italy): the Recovery of Reef Communities after the Permian/Triassic Crisis. Facies 28: 181–256.

Reviewer 3 ·

Basic reporting

no comment

Experimental design

This study bring much needed paleoecological data from the western Panthalassa region to the discussion of marine recovery following the EPME. The authors are thorough in their quantitative assessment of recovery patterns and contributes significantly to comparisons between the fauna in northeastern Japan, North American and central European. The newly discovered ammonoid taxa are a significant contribution to the biostratigraphy of the region. The experimental design is solid, and the quantitative methods are standard and well implemented. The ammonoid systematics appears thorough, though this is not my area.

The ichnotaxa data is the most compelling for both of the authors’ main interpretations: a state of semi-recovery following the end of the Smithian thermal event in this region, and an exploration of the presence of absence of a ‘habitable zone’ here (further discussion below). The body fossil data is less compelling, mostly due to the low taxonomic resolution. The non-ammonoid data does not contribute much to the authors’ main conclusions (excluding the Eumorphotis size data), so I am tempted to say that is it not even needed in the manuscript, since the ichno data is so abundant in comparison.

Validity of the findings

In the discussion of the Habitable Zone hypothesis, it is not clear to me from the authors’ explanation as to how the patterns that they are observing are specific to the idea of the Early Triassic habitable zone, and not just ichnotaxon distribution patterns that are expected in a normal onshore-offshore setting during any other time (as the authors partially address). The authors still observe relatively high ichnotaxon diversity in the deeper (lower Hd and He) facies, and the geochemical proxies indicate that it is likely still weakly oxygenated in those settings, at least enough to support some relatively complex ichnotaxa like Rhizocorallium. The shallowest facies show the lowest diversity, interpreted to be from sediment influx and wave energy stress, but how is this unique to the habitable zone hypothesis if a discussion of SST/thermal stress isn’t also incorporated into this interpretation?

Additional comments

Figure 4: The delineation for ‘lower shoreface’, ‘offshore transition’, and ‘distal offshore’ at the top of the figure does not line up with the definition of the zones based on FWWB and SWB. For example, the ‘offshore transition’ zone is defined as and should be shown to start at FWWB and end at SWB.

Figure 5A: The lower center area of the specimen where the sutures are visible is out of focus in the photo. I recommend that the authors try to retake the photo, if possible, using z-stacking software (freeware available). This may be out of the authors’ control, but the images would benefit from increasing contrast and decreasing brightness in a program like Photoshop ore something similar.

---

## Round 0.2 · accepted · Accept

Thank you for meticulously addressing our suggestions which has made the manuscript even easier to follow and of broader relevance. I did not find any additional points to raise in the revised manuscript. I agree with the reviewers' assessment that the manuscript can be accepted as is. I look forward to seeing it published.

·

Basic reporting

The authors have accounted thoroughly for my comments. Looking forward to see this paper published!

Experimental design

fine as it is

Validity of the findings

fine as it is

Reviewer 3 ·

Basic reporting

I find that the authors have satisfactorily addressed most of the comments made by the three reviewers, and where they did not amend the text, offered an acceptable counterargument.

Experimental design

No significant changes were made to the experimental design. The systematics section was improved.

Validity of the findings

There were no significant changed to the study's findings. In response to reviewer comments, the authors have clarified certain discussion points.